# The Fungal Root Endophyte *Serendipita indica* (*Piriformospora indica*) Enhances Bread and Durum Wheat Performance under Boron Toxicity at Both Vegetative and Generative Stages of Development through Mechanisms Unrelated to Mineral Homeostasis

**DOI:** 10.3390/biology12081098

**Published:** 2023-08-07

**Authors:** Ali Kaval, Halil Yılmaz, Sedef Tunca Gedik, Bahar Yıldız Kutman, Ümit Barış Kutman

**Affiliations:** 1Institute of Biotechnology, Gebze Technical University, Gebze 41400, Kocaeli, Türkiyebykutman@gtu.edu.tr (B.Y.K.); 2Department of Molecular Biology and Genetics, Gebze Technical University, Gebze 41400, Kocaeli, Türkiye; halil.yilmaz@gtu.edu.tr (H.Y.); sgedik@gtu.edu.tr (S.T.G.)

**Keywords:** *Serendipita indica*, abiotic stress, boron toxicity, wheat, endophytic fungi, plant-microbe interactions, membrane damage, grain yield

## Abstract

**Simple Summary:**

Boron is an essential micronutrient for plants; however, boron toxicity has long been identified as a common nutritional problem on a global scale and is attracting increasing interest. Excessive concentrations of B significantly reduce crop yield, including that of wheat, which is a staple food throughout the world. Wheat cultivation is particularly common in arid and semi-arid regions where climate change and irrigation can aggravate the risk and severity of B toxicity. *Serendipita indica* is a versatile root endophytic fungus. Studies show that plants colonized by *Serendipita indica* exhibit a wide range of beneficial effects, encompassing enhanced growth and yield, higher tolerance to various biotic and abiotic stresses, and improved water and nutrient balance. The present study revealed that under B toxicity, the growth, membrane integrity, tissue P content, and yield of wheat can be significantly improved by this endophyte without reducing toxic B concentrations in the shoot. Considering that *Serendipita indica* can be axenically produced on a large scale, this endophyte can be used as a microbiological agent for promoting sustainable agriculture under adverse soil conditions like boron toxicity. Further studies are needed to evaluate the potential and specific mechanisms of *Serendipita indica* under field conditions.

**Abstract:**

While the importance of beneficial soil microorganisms for soil health and crop performance has been receiving ever-increasing attention, *Serendipita indica* has been widely studied as a fungal root endophyte with significant potential for increasing the stress tolerance of host plants. Boron (B) toxicity as an adverse soil condition is particularly prevalent in arid and semi-arid regions and threatens crop production. Studies on *S. indica*-wheat symbiosis are limited, and effects of *S. indica* on crops have never been reported in the context of B toxicity. Here, two pot experiments were conducted under greenhouse conditions to investigate the effects of *S. indica* on the growth and yield parameters of bread (*Triticum aestivum*) and durum wheat (*Triticum durum*) grown at different levels of B toxicity in native vs. sterilized soil, and parameters related to root colonization, membrane damage, oxidative stress, chlorophyll, and mineral nutrition were measured to elucidate the physiological mechanisms of damage and benefit. Boron toxicity decreased early vegetative growth and grain yield, but it did not affect the straw dry weight of mature plants, whereas *S. indica* significantly enhanced the vegetative growth, straw dry weight, and the grain number of both wheat species. Membrane damage as demonstrated by increased lipid peroxidation and relative electrolyte leakage was caused by B toxicity and alleviated by *S. indica*. The benefits provided by *S. indica* could not be attributed to any significant changes in tissue concentrations of B or other minerals such as phosphorus. Soil sterilization generally improved plant performance but it did not consistently strengthen or weaken the effects of *S. indica*. The presented results suggest that *S. indica* may be used as an effective microbial inoculant to enhance wheat growth under adverse soil conditions such as B toxicity through mechanisms that are possibly unrelated to mineral homeostasis.

## 1. Introduction

Boron (B) is required for the normal growth and development of plants and has been considered an essential micronutrient since 1923 [1,2,3]. Although B deficiency is one of the most widespread micronutrient deficiencies in soils globally [2,4,5,6], B toxicity is also a common adverse soil condition and thus a common abiotic stress factor, which is gaining increased attention in different countries, including Türkiye, Australia, Russia, Italy, Egypt, Iraq, Libya, Jordan, Syria, Morocco, India, Chili, and California, among others [7,8,9,10]. In the Central Anatolian Plateau of Türkiye, B toxicity has been considered one of the two most prominent micronutrient problems, along with zinc deficiency [11,12,13].

Boron toxicity in Angiosperms leads to various structural, metabolic, and physiological consequences, including, but not limited to, loss of membrane function and integrity, impaired cell wall structure, changes in metabolism, reduced mitotic activity in root meristems, lower levels of suberin and lignin, and reductions in chlorophyll contents and photosynthetic rates [14,15,16,17]. Visible symptoms of B toxicity include diminished growth of shoots and roots [18], chlorosis (i.e., tip and/or marginal) and necrosis [19,20,21]. The link between B toxicity and oxidative stress has been investigated in different crop species. Reactive oxygen species (ROS) are produced excessively under oxidative stress resulting in membrane damage and ultimately cell death [22,23].

It was shown that a strong anti-oxidative response against oxidative stress caused by toxic levels of B may alleviate the damage in various plants [22,23,24,25,26,27]. In different cereals, including maize and wheat, B toxicity was associated with elevated H_2_O_2_ and MDA concentrations and increased electrolyte leakage [26,28]. However, in a study on barley, the increases in MDA levels and electrolyte leakage which were caused by B toxicity did not correlate with H_2_O_2_ concentrations or anti-oxidative enzyme activities, and it was concluded that membrane damage by B toxicity was not due to reactive oxygen species [29].

In wheat and barley, the extent of the effects of B toxicity on yield and the visual symptoms of B toxicity differed markedly among genotypes [13,30,31,32,33,34]. It was reported that excessive B restricts the production of starch or creates B-carbohydrate complexes, thereby causing delayed grain formation [35]. Because the range between critical concentrations for B deficiency and B toxicity is rather narrow, the yield and quality of many crops are often affected either by its deficiency or toxicity [36,37]. Boron deficiency can relatively easily be prevented or corrected via proper soil and/or foliar fertilization applications, whereas B toxicity has been more challenging to manage so far. Soil amendments that modulate soil pH and, thus, decrease the availability of B or facilitate its leaching can be used to alleviate B toxicity with limited success [38]. In this context, beneficial soil microorganisms can offer much more cost-effective, eco-friendly, and sustainable solutions.

*Serendipita indica* (syn. *Piriformospora indica*) is a root colonizing fungus (endophytic) belonging to the order Sebacinales under Basidiomycetes [39,40,41,42]. It can be cultured axenically and able to grow on a wide range of media [43,44,45]. Although *S. indica* is not an arbuscular mycorrhiza (AM) species, it shares some functional, morphological, and growth-promoting characteristics with AM fungi [46,47,48]. In symbiosis with *S. indica*, host plants gain a wide range of benefits, such as enhancement of growth [49,50], water and nutrient uptake [51], photosynthetic activity and seed production [52], plant tolerance to major biotic and abiotic stresses [53,54,55,56], and adventitious root formation in cuttings [46]. Hitherto, studies have reported *S. indica*-induced enhancement of stress tolerance for numerous crops including wheat [57,58,59], maize [60,61], barley [62], rice [63], and tomato [54]. The ability of *S. indica* to extract, mobilize, and transport macronutrients (i.e., P and N), as well as micronutrients (i.e., Fe, Zn, Mn, Cu) from the soil has been demonstrated as a mechanism contributing to its beneficial effects on host plants [64,65,66,67]. Moreover, under biotic and abiotic stress conditions, *S. indica* has the capability to improve plant vigor through altering the antioxidative system and enhancing the detoxification of ROS [61,62,68,69].

The effects of B toxicity on wheat include stunting, impaired shoot, and root growth, delayed development, visual leaf symptoms, such as necrosis, and most importantly, yield losses [31,70,71,72]. The proposed primary mechanisms responsible for B tolerance in wheat include exclusion from roots, diminished translocation to shoots and avoidance by virtue of shallow root structures [73,74]. Moreover, studies revealed that tolerant cultivars typically had lower shoot B concentrations, suggesting that exclusion from roots and/or decreased translocation to shoots may have a more crucial role than tissue tolerance in this context [31,75]. A study conducted on durum wheat (*Triticum durum* Desf.) under excessive B concentrations showed that the AM fungus *Glomus clarum* alleviated B toxicity symptoms and reduced shoot and root B concentrations without having an effect on biomass [76]. The rhizobacterium *Bacillus pumilus* alleviated the effects of B toxicity in tomato (*Lycopersicum esculentum* L.) and rice (*Oryza sativa* L.) [77,78]. Mycorrhiza-inoculated Carrizo citrange seedlings showed decreased B concentrations and milder symptoms than the non-mycorrhizal plants when irrigation water contained toxic concentrations of B [79]. According to Liu et al., AM-inoculated *Puccinellia tenuiflora* plants had improved biomass and lower tissue B concentrations under combined stresses of B toxicity, salinity, and drought [80].

The potential benefits of *S. indica* have never been addressed in the context of B toxicity in wheat or any other crop species. To the best of our knowledge, there is also no published study that compares the growth-promoting and stress-alleviating effects of *S. indica* on bread and durum wheat cultivars. In this study, a spring bread wheat (*Triticum aestivum* L., cv. Nusrat) and a durum wheat (*Triticum durum* Desf., cv. Saricanak-98) cultivars were grown in the presence or absence of *S. indica* under greenhouse conditions in native or autoclaved soil supplemented with different concentrations of B. Although there is no study in the literature comparing specifically these two cultivars in terms of B tolerance, it is generally accepted that durum wheat is more susceptible to B toxicity than bread wheat [7,81]. In the experiments reported herein, various parameters (growth, grain yield and stress, as well as mineral nutrition and partitioning) were measured to comparatively document the responses of these two wheat species to *S. indica*, depending on B toxicity, and to gain insight into the physiological and nutritional mechanisms of the observed effects.

## 2. Materials and Methods

In this study, first soil experiment was conducted under greenhouse conditions to investigate the physiological effects of *S. indica* on the vegetative growth and B toxicity tolerance of bread (*Triticum aestivum* L., cv. Nusrat) and durum wheat (*Triticum durum* Desf., cv. Saricanak-98) in sterilized vs. non-sterilized soils. Then, a second soil experiment was performed to examine the effects of *S. indica* on the yield components and mineral balance of wheat in the absence or presence of B toxicity, depending on the sterilization of the soil.

### 2.1. Greenhouse and Growth Chamber Conditions

Soil experiments were conducted in the Research and Development Greenhouse at the Institute of Biotechnology of Gebze Technical University, Kocaeli, Turkiye. The greenhouse used in this study is a Venlo-type glasshouse located in Gebze at the following geographic coordinates: 40°48′40″ N, 29°21′29″ E. It has automated climate control (evaporative cooling and heating) and complementary lighting. The daytime temperature was retained at 24 ± 3 °C and the nighttime temperature at 20 ± 3 °C in the greenhouse throughout the experiments.

### 2.2. Plant Germination

Durum and bread wheat seeds were germinated in perlite moisturized with 4 mM N in the form of Ca(NO_3_)_2_.4H_2_O solution for 5 days under controlled climatic conditions (light/dark periods: 16/8 h; temperature (light/dark): 25 °C/20 °C; relative humidity (light/dark): 60%/70%; photosynthetic flux density: 400 μmol m^−2^ s^−1^) before being transferred to soil.

### 2.3. S. indica Growth Condition and Soil Application

After providing *Serendipita indica* (DSM11827), the chlamydospores were preserved in glycerin at −80 °C. For the experiments, the fungus was maintained in axenic culture and freshly re-isolated from *Arabidopsis thaliana* roots, as described by Johnson et al. [82]. The fungus was cultured in a modified Kaefer agar medium according to Hill and Kaefer [83], incubated at 28 °C in the dark for 1 month, and stored at +4 °C. In order to obtain the fungus for soil inoculation, 4 fungal plugs (1 cm diameter) were removed from the edge of active colonies and used to inoculate Erlenmeyer flasks, each containing 200 mL of modified Kaefer liquid medium, composed of: peptone 2 g·L^−1^, casein hydrolysate 1 g·L^−1^, yeast extract 1 g·L^−1^, D-glucose 20 g·L^−1^, 50 mL·L^−1^ of macronutrients solution (NaNO_3_ 12 g·L^−1^, KCl 10.4 g·L^−1^, MgSO_4_.7H_2_O 10.4 g·L^−1^, and KH_2_PO_4_ 30.4 g·L^−1^), 10 mL·L^−1^ of micronutrients solution [H_3_BO_3_ 1.1 g·L^−1^, MnSO_4_.H_2_O 0.37 g·L^−1^, ZnSO_4_.7H_2_O 2.2 g·L^−1^, CuSO_4_.5H_2_O 0.16 g·L^−1^, (NH_4_)_6_Mo_7_O_24_.4H_2_O 0.11 g·L^−1^, CoCl_2_.6H_2_O 0.17 g·L^−1^], 1 mL·L^−1^ of Fe-EDTA solution (17.2 g·L^−1^), and agar 10 g·L^−1^. Medium pH was adjusted to 6.5 with KOH (1 N) prior to sterilization (at 121 °C for 20 min). Finally, 1 mL·L^−1^ of vitamin solution containing thiamin 100 g·L^−1^, glycine 0.4 g·L^−1^, nicotinic acid 0.1 g·L^−1^, and pyridoxine 0.1 g·L^−1^ were added. The fungus was incubated at 28 °C and 180 rpm for 14–21 days in an orbital shaking incubator (ZWYR-D2401, LABWIT Scientific, Burwood East, Australia). Fresh cultures were routinely grown on liquid Kaefer medium for the soil experiments [83,84,85]. For fungus harvest, liquid cultures were centrifuged at 7500 rpm and 20 °C for 10 min, and the supernatant was discharged to completely remove the medium. The pellets were pooled and resuspended in dH_2_O. The suspension was stirred continuously while aliquots were applied to pots.

### 2.4. Quantification of Spore Density Using Hemocytometer

The quantification of pear-shaped spores was carried out according to Kumar et al. [86], and Singhal et al. [87]. Pear-shaped chlamydospores of *S. indica* were attached to the mycelium. The spores were dislodged by adding 200 µL of ‘Tween 80’ (2%) solution to 10 mL of the liquid culture, and vigorously vortexed for 2 min. Afterwards, the mixture was sonicated at 30% amplitude for 20 s on and 30 s off cycle on dry ice for a total of 5–8 min. After detachment of the spores, 2 mL of sonicated culture was transferred to Eppendorf (main stock) and gently vortexed. Prior to the quantification of spores by using a hemacytometer counting chamber (Neubauer improved, Isolab, Eschau, Germany), successive dilutions of the main stock were carried out to monitor and count the spores.

### 2.5. Experimental Design

The experimental soil was transported from a field in Tuzla, Istanbul. The field had not been in use for at least 3 years. It was air-dried and then stored in bags in a dry storage room. It had clay-loam texture, low CaCO_3_ (0.3%) and low organic matter content (0.9%). It was non-saline soil with a slightly alkaline pH (7.67).

The design included soil sterilization as a factor because the native soil may potentially harbor soil-borne pathogens, as well as non-pathogenic but competitive microorganisms which may positively or negatively interfere with *S. indica* survival, root colonization and function. One half of the soil was autoclaved at 121 °C for 45 min while the other half was kept in its native state. Similarly, one half of the perlite was autoclaved at 121 °C for 20 min. The autoclaved substrates were air-dried under greenhouse conditions. Pots were filled with 2.1 kg of either non-autoclaved or autoclaved soil. Before the seedlings were transplanted from perlite to soil, the following mineral nutrients were added to each pot as concentrated solutions and uniformly incorporated into the soil by mixing thoroughly (per kg air-dry soil): 250 mg N in the form of Ca(NO_3_)_2_.4H_2_O; 100 mg P in the form of KH_2_PO_4_; 10 mg Cl in the form of KCl. To loosen up the soil structure, 150 g of either non-autoclaved or autoclaved perlite were added to pots and then mixed thoroughly. Transfer of 5-day-old seedlings to the pots and *S. indica* inoculation were carried out as follows: approximately 250 mL of the soil–perlite mixture from the top was taken and the surface was drenched with 100 mL of inoculum, containing 16.8 g of mycelium-spore mixture with a final spore concentration of 9.65 × 10^5^ per mL. In the control (non-inoculated) group, pots were drenched with the same volume of de-ionized water in the same way. The seedlings were diligently placed on the surface (10 seedlings per pot). The 250 mL soil–perlite mixture was filled back into the pots to cover all the roots. Finally, the top was drenched again with the same volume of inoculum.

The first soil experiment comprised a fully factorial design with 4 pot replicates per treatment (2 inoculation, 2 soil sterilization, 2 wheat cultivars, 4 B toxicity levels, 128 pots in total) and the pots were completely randomized in the greenhouse. Saricanak-98 and Nusrat were grown with or without *S. indica* inoculation in either autoclaved or non-autoclaved soil at 4 B levels [1 mg (control), 10 mg (low), 20 mg (medium) and 30 mg (high) B per kg dry soil in the form of boric acid]. Plants were irrigated with dH_2_O water every day or on alternate days throughout the experiment depending on the seasonal conditions and the biomass of the plants. Pots were manually weeded when needed. The seedlings were thinned to 8 per pot after 14 days. The experiment was terminated by harvesting the shoots and roots separately at 41 days after sowing (DAS).

In the second experiment, there were 3 B levels [per kg air-dry soil: 1 mg (control), 10 mg (medium) and 20 mg (high)]. Based on the results of the first experiment, the 30 mg B per kg soil level was excluded from this experiment because of the high degree of toxicity which may be irrelevant for practical applications. The basal fertilization was conducted in exactly the same way as in the first experiment but to meet the higher N demand of the plants to grow to full maturity, 45 DAS, the pots were supplied with an additional 150 mg N in the form of Ca(NO_3_)_2_.4H_2_O. The experimental design and procedure were otherwise the same as described above for the first experiment. The second experiment had a fully factorial design with 4 pot replicates per treatment, (2 inoculation, 2 soil sterilization, 2 wheat cultivars, 3 B toxicity levels, 96 pots in total), and the pots were completely randomized in the greenhouse. Here, for inoculation, 14 g of mycelium-spore mixture containing 13.8 × 10^5^ spores per ml was added to each pot. The seedlings were thinned to 5 per pot 14 DAS. The experiment was terminated when grains reached full maturity and the plants were fully senescent (at least 80%, 111 DAS). The spikes and the straw were harvested separately, and the straw samples were dried at 60 °C for 48 h. Grains were separated from husk by using a thresher to determine the grain yield per plant and the number of grains per plant. Then, the husk samples were combined with the corresponding straw samples. The grain and straw samples were acid-digested, and the concentration of B, P, and Mn were determined by inductively coupled plasma optical emission spectrometry (ICP-OES, Agilent, CA, USA), as described in “Acid Digestion and Mineral Analyses”.

### 2.6. Dry Weight Determination

In the first soil experiment, plants were harvested by cutting them 1–2 cm above the soil level. Then, 4 out of 8 seedlings were harvested, rinsed in dH_2_O and dried at 70 °C for 48 h. Before weighing, the samples were kept at room temperature for 2 h. For the second soil experiment, the spikes and the straw were harvested separately. The straw samples were dried at 70 °C for 48 h to make sure that the any remaining moisture was removed before dry weight measurement.

### 2.7. Relative Electrolyte Leakage (REL)

As a measure of membrane integrity, REL of the second youngest leaves from each pot was determined using a slightly modified version of the method described by Tyagi et al. [88], and Lutts et al. [89]. Accordingly, leaves were cut and washed under running dH_2_O. In closed vials, leaf segments with the same length were completely submerged in 50 mL of dH_2_O for 6 h at 24–25 °C, and then, the electrical conductivity (EC_1_) was measured with an EC meter (Mettler Toledo, Greifensee, Switzerland). After boiling the samples at 95 °C for 1 h, the electrical conductivity (EC_2_) was measured again. REL was calculated as follows: REL (%) = (EC_1_/EC_2_) × 100.

### 2.8. Chlorophyll Determination

The second fully expanded leaves were freshly used from each pot to determine the total chlorophyll content spectrophotometrically according to the methods described by Lichtenthaler [90], and Abadi and Sepehri [58]. After a fine chopping, portions weighing approximately 0.5 g were measured off on an analytlical balance. Fresh leaf samples were frozen by using liquid nitrogen and kept at −80 °C, subsequently. Homogenization of frozen leaf specimens were performed by using a ball mill (MM200; Retsch, Haan, Germany), liquid nitrogen and 5 mL of 80% acetone. In this way, a primary acetone extract including all chloroplast pigments was provided. The extract was then centrifuged twice (7500 and 12,000 rpm for 15 min at 4 °C, respectively). The supernatant was diluted by adding 5 mL of 80% acetone per ml since the concentration of pigments was too high for reading to be applied on a spectrophotometer in the most cases. The measurements were carried out by using a UV-Visible spectrophotometer (Cary 300 Bio; Varian, Australia).

### 2.9. Lipid Peroxidation Assay

Lipid peroxidation was determined spectrophotometrically using a modified thiobarbituric acid–malondialdehyde (TBA-MDA) assay, according to Heath and Packer [91]. To execute the analysis, the third and fourth fully expanded leaves were freshly used from each pot. Fresh leaf samples (ca. 0.5 g) were removed from liquid nitrogen and quickly ground using a liquid nitrogen-cooled ball mill (MM200; Retsch, Haan, Germany) at a speed of 30 oscillations per second for 30 s. The homogenization was repeated after adding 12.5 mL of 80% ethyl alcohol. The leaf homogenates then were clarified by centrifugation (at 4 °C for 15 min and at 15,000× *g*), and aliquots of the supernatant were stored at −80 °C. The assay was carried out in two stages. At the first stage, after adding 1 mL of 20% trichloroacetic acid (TCA) containing 1 mL of 0.01% butylated hydroxytoluene (BHT) to 1 m Laliquot of the supernatant, the mixture was vortexed vigorously and heated at 95 °C for 30 min. Afterwards, the mixture was quickly cooled in an ice-bath and centrifuged at 10,000× *g* for 10 min. At the second stage, after adding 1 mL of 20% trichloroacetic acid (TCA) containing 0.65% thiobarbituric acid (TBA) to 1 mL aliquot of the supernatant, remaining steps were carried out to the mixture as explained in the first stage. Absorbances were read at 440 nm, 532 nm, and 600 nm and corrections were made for interfering compounds.

### 2.10. Acid Digestion and Mineral Analyses

About 0.2 g of dried and ground samples were weighed into the digestion vessels. Then, 2 mL of 30% H_2_O_2_ and 5 mL of 65% HNO_3_ were added to the vessels. Acid digestion was conducted in a closed-vessel microwave system (MarsExpress; CEM Corp.; Matthews, NC, USA). Then, the vessels were removed and vented in a fume hood after cooling to room temperature. The total sample volume was finalized up to 20 mL with ddH_2_O and filtered through quantitative filter paper. The concentrations of all essential macro- and micronutrients, except N, were measured by ICP-OES in digested samples. Values for B, P and Mn were reported in the present manuscript. Certified standard reference materials (National Institute of Standards and Technology, Gaithersburg, MD, USA) were used as control. Certified calibration solutions were used for instrument calibration (TraceCERT^®^, Sigma-Aldrich, Darmstadt, Germany).

### 2.11. Histochemical Analysis

The root staining procedure was carried out according to the method by Kumar et al. [92] with some modifications. To demonstrate colonization, 10 root samples were selected randomly from colonized and non-colonized wheat roots. The root samples were harvested randomly from corresponding treatments (10 root samples from each pot, 320 samples in total), washed thoroughly under running tap water and cut into 3 cm segments from the tips. The root segments were immersed in 10% KOH (*w*/*v*) for 15 min (or until become transparent) to soften, acidified with 1 M HCl solution for 10 min, and ultimately stained with 0.02% Trypan blue solution overnight [93,94]. The following day, the segments were rinsed with dH_2_O to remove excess dye and de-stained with 50% lactoglycerol (lactic acid: glycerol: dH_2_O, 1:1:1, *v*/*v*/*v*) for 1–2 h prior to observation using a fluorescent-light microscope (Olympus BX-51). All root segments were photographed with KAMERAM software, assisted by a KAMERAM fluorescent camera.

### 2.12. Genomic DNA Isolation from Plant Roots and PCR Analysis

The presence of *S. indica* in wheat roots was shown by PCR targeting the specific EF-1-alpha (*tef*) gene (AJ249912). The genomic DNA of *S. indica* was isolated from wheat roots by using a modified CTAB (cetyltrimethylammonium bromide) method [95,96,97]. PCR reactions were executed in a final volume of 50 µL consisting of 10 µL of 5× MyTaq reaction buffer (Bioline), 1 µL of each primer (20 µM), 1 µL of MyTaq DNA polymerase (500U, Bioline MyTaq™ DNA Polymerase), and 2 µL of genomic DNA (ca. 100 ng). EF-1-alpha (*tef*) gene (227 bp) was amplified by using *PITEF* primer pairs [Pitef-F (5′-TCGTCGCTGTCAACAAGATG-3′) and Pitef-R (5′-GAGGGCTCGAGCATGTTGT-3′)] which were synthesized according to the sequence available in GenBank. PCR reactions were carried out in a T-Professional Thermocycler (Biometra, Gottingen, Germany) set to the following reaction conditions: an initial denaturation at 94 °C for 5 min, followed by 35 cycles of 40 s at 94 °C, 40 s at 59 °C (annealing) and 30 s at 72 °C (extension). PCR reactions without DNA template served as negative control. PCR products were run in 0.8% agarose gel (GeneON Bioscience, Ludwigshafen, Germany) with 1× TAE buffer (ClearBand TAE 50× Electrophoresis Buffer) at 90 V for 40 min. The approximate size of DNA fragments was determined by using 100 bp DNA ladder (GeneRuler 100 bp DNA Ladder, Thermo Fischer Scientific, Waltham, MA, USA). Agarose gel was stained with a nucleic acid staining solution (RedSafe™) prior to observation under UV illuminator with Azure c600 Gel Imaging System (Azure Biosystems, Dublin, CA, USA).

### 2.13. Calculations and Statistical Analysis

Statistical analysis was carried out using statistical software (JMP, version 16.0.0). All reported values represent means of 4 replicates. The significance of the effects of the treatments and their interactions on the traits of interest was evaluated by analysis of variance (ANOVA). Additionally, significant differences between means were determined by Tukey’s ‘protected’ honestly significant difference (HSD) test at 95% confidence (*p* ≤ 0.05) only where ANOVA reported a significant effect.

## 3. Results

Histological studies and characterization of fungal colonization were carried out by observing inter- and intra-cellular pear-shaped chlamydospore formations (Figure 1B,C) in roots colonized with *S. indica* at medium B level as compared with corresponding non-colonized (control) plants (Figure 1A). The chlamydospores were scattered from one another or grouped in chains with two-to-many spores. Not all spores displayed the characteristic structure but some of them also appear to be round, ovoid, or rectangular in shape (Figure 1B,C).

In addition to the histological studies, the presence of *S. indica* in wheat roots was also proven with PCR by amplifying the *tef* gene (Figure 2). *tef* DNA fragment of *S. indica* was detectable at 41 DAS in the roots of inoculated plants. Moreover, non-inoculated wheat roots showed no band formation as expected. Taken together, experimental plants of both cultivars were successfully infected and colonized by *S. indica* and the fungus persisted in host tissues, independently of the B supply.

The general condition and visual appearance of 41-day-old plants varied greatly according to the treatments at this stage. Under non-autoclaved soil conditions, increasing the B supply to toxic levels impaired plant vigor, and caused leaf symptoms of increasing severity, encompassing chlorosis and necrosis for both cultivars while inoculated plants exhibited improved vigor (Appendix A).

As displayed by three-way ANOVA, the triple interaction of B supply, soil sterilization, and inoculation had a significant effect on the shoot dry matter of Nusrat but not on that of Saricanak-98 (Table 1). However, for both cultivars, the main effects of all three factors on the shoot dry weight were significant (Table 1). Increasing the B supply to toxic levels reduced the shoot dry weight in a dose-dependent manner in Nusrat, as well as Saricanak-98 (Table 2). In the presence of *S. indica*, the shoot dry weights of plants were higher than those grown without *S. indica* for both Nusrat and Saricanak-98. Moreover, plants grown in sterilized soils had higher shoot dry weights than those grown in non-sterilized soils. Overall, the shoot dry matter production was positively affected by *S. indica* and soil sterilization (Table 2).

It was observed that *S. indica* inoculated plants of both cultivars showed decreased REL (Table 1; Figure 3A). Soil sterilization was associated with increased REL in Saricanak-98, (Figure 3B). The effect of B toxicity on REL was significant at medium and high B levels for both cultivars whereas the low level of B toxicity did not result in significant increases in REL (Figure 3C).

The shoot MDA concentration was significantly affected by all single treatments in Nusrat; however, in Saricanak-98, it was affected by only inoculation and B supply (Table 1). Inoculated plants displayed considerably lower shoot MDA concentrations in both cultivars (Figure 4A). Even though soil sterilization reduced shoot MDA concentrations in Nusrat, it did not have any effect in Saricanak-98 (Figure 4B). Increasing levels of B toxicity generally caused significant increases in MDA concentrations; however, the lowest level of toxic B supply was not associated with a significant increase with respect to the control B supply (Figure 4C).

The total chlorophyll concentration, in Saricanak-98, was affected by inoculation and B supply, as well as their double interaction; however, in Nusrat, only soil sterilization and its double interactions with inoculation and B supply significantly affected leaf chlorophyll levels (Table 1). According to the collective data in Table 3, in durum wheat, improved chlorophyll levels were measured in the inoculated group. Although soil sterilization enhanced the total chlorophyll in Nusrat, it did not in Saricanak-98. Increasing the B supply did not result in any significant difference in the leaf chlorophyll concentrations in bread wheat while in durum wheat, no consistent response to increasing B levels was observed (Table 3).

According to ANOVA, the shoot B concentration was significantly affected by the single effects of soil sterilization and B supply as well as their double interaction in Nusrat. In Saricanak-98, only the main effects of soil sterilization and B supply on the shoot B concentrations were significant (Table 4). The presence of *S. indica* did not cause any positive or negative effect on shoot B concentrations in either cultivar (Figure 5A). However, soil sterilization reduced the shoot B concentrations in both the bread and durum wheat cultivars (Figure 5B). Increasing the B supply from the control to the high level resulted in a dose-dependent elevation of the shoot B concentrations for both cultivars (Figure 5C).

The shoot Mn concentration was significantly affected by all treatments and the double interaction of soil sterilization and B supply in Nusrat (Table 4). In Saricanak-98, however, it was significantly affected by just soil sterilization and its double interactions with inoculation and B supply. Even though *S. indica* reduced the shoot Mn accumulation in Nusrat, it did not result in any considerable change in Saricanak-98 (Figure 6A). Soil sterilization dramatically increased (almost 3X) the measured shoot Mn concentrations (Figure 6B). The presence or absence of significant effects of B supply on the shoot Mn concentrations depended on the cultivar (Figure 6C).

The main effects of all treatments and the effect of the interaction of soil sterilization and B supply on the shoot P concentrations were significant (Table 4). Although inoculated bread wheat plants showed reduced shoot P concentrations than non-inoculated ones, this effect was not observed for durum wheat plants (Figure 7A). In addition to that, soil sterilization increased the shoot P concentrations in both cultivars (Figure 7B). Experimental plants in the high B treatment group exhibited the highest shoot P concentrations while those in the control B group showed the lowest P concentrations in both cultivars (Figure 7C).

In the second experiment, where the plants were grown to full maturity, the general condition and visual appearance of the plants varied greatly according to the treatments at 50 DAS as shown in Figure 8. In non-autoclaved soil, increasing the B level from control to medium resulted in typical leaf symptoms, including chlorosis and necrosis starting from the tips of mature leaves. In Saricanak-98, which was ahead of Nusrat at 50 DAS in terms of the developmental stage, B toxicity also resulted in impaired stem elongation (Figure 8C,D). Inoculation with *S. indica* improved the plant vigor apparently in both the bread and durum wheat cultivars (Figure 8).

Analysis of variance revealed that the grain number was significantly affected by all treatments in both cultivars (Table 5). Inoculation with *S. indica*, as well as soil sterilization resulted in significantly higher grain numbers in both cultivars whereas the grain number was reduced by increasing B supply in a dose-dependent manner (Figure 9A–C).

The straw dry matter of mature plants was significantly affected by both inoculation and soil sterilization (Table 5). Inoculated plants had significantly higher straw dry weights (Figure 10A). Similarly, plants grown in sterilized soil had higher straw biomass than those grown in non-sterilized one (Figure 10B). In response to B supply, the straw dry weights of the tested cultivars did not change (Figure 10C).

Both soil sterilization and B supply significantly affected the grain yield of the tested bread and durum wheat cultivars (Table 5). Inoculation with *S. indica* failed to improve the grain yield per plant (Figure 11A) while soil sterilization was associated with higher yields (Figure 11B). Increasing levels of B toxicity resulted in dose-dependent yield losses in Nusrat; however, in the case of Saricanak-98, only the highest level of B caused a significant decrease in grain yield (Figure 11C).

The grain B concentration was significantly affected by only B supply and its interaction with soil sterilization in Nusrat; however, in Saricanak-98, it was significantly affected by all factors as well as the interaction of soil sterilization with B supply (Table 5). According to the collective data at Table 6, even though the grain B concentration was reduced by *S. indica* in Saricanak-98, it was not affected in Nusrat. Experimental plants grown in autoclaved soil exhibited lower grain B concentrations in the case of Saricanak-98. Increasing B supply resulted in dose-dependent increases in the grain B concentrations for both cultivars (Table 6).

In contrast to the effects observed for the grain B concentration, the straw B concentration was reduced by *S. indica* in Nusrat but not in Saricanak-98 (Table 4 and Table 7). Soil sterilization resulted in reduced straw B concentrations in both cultivars whereas increasing B supply caused dose-dependent increase in straw B levels (Table 7).

## 4. Discussion

Although *Serendipita indica* has great potential as a plant growth-promoting micro-organism, practical applications of this endophyte in crop production require the elucidation of its crop-specific physiological effects. Root colonization with *S. indica* may be particularly beneficial under stress conditions but its potential in the context of B toxicity has not been documented before.

Decreases in growth, yield, and/or quality are the most practically relevant consequences of stress in crop plants. Plants exposed to toxic levels of B typically display impaired growth of shoots and roots [18,98,99]. *S. indica* inoculation was documented to have a positive effect on the biomass production of a broad variety of crops [45,53,100,101,102,103]. Here, to study the potential contribution of *S. indica* to the growth and yield of wheat in the absence or presence of B toxicity, wheat plants were inoculated with the fungus. The current study revealed that shoot dry matter of inoculated plants were significantly higher than non-inoculated plants regardless of B toxicity (Table 2, Appendix A).

Even though, to date, the potential benefits of *S. indica* have never been reported in the context of B toxicity in wheat, numerous studies have been conducted on wheat to study the protective effects of *S. indica* under different abiotic or biotic stress conditions. For instance, under Cd toxicity, *S. indica* colonized wheat plants showed improved growth and chlorophyll content compared to the control [103]. Moreover, in that study, the Cd concentration was reduced in shoot in the presence of *S. indica*. A study conducted on wheat plants showed that *S. indica* could protect the plants from crown rot caused by *Fusarium* at the seedling stage [104]. According to Zarea et al. [105], *S. indica* inoculation led to improved wheat (*Triticum aestivum* L.) growth in both the absence and presence of salinity, and symbiosis was associated with enhanced salinity tolerance. In agreement with these studies, the results reported herein showed that both bread and durum wheat responded positively to symbiosis with *S. indica* under both control and stress conditions in terms of growth (Table 2 and Appendix A). While, to the best of our knowledge, all previously published studies on wheat and *S. indica* investigated specifically bread wheat in this context, bread and durum wheat plants were compared here in terms of their responsiveness to this symbiosis. According to the results presented in Table 2, durum wheat benefited more from this interaction than bread wheat.

Plant growth can be enhanced by soil sterilization through autoclaving or other techniques thanks to the elimination of soil-borne pathogens and competing micro-organisms [106,107,108]. In the present investigation, it appeared that soil sterilization resulted in significantly elevated shoot dry matters irrespective of *S. indica*, which is in agreement with the literature. To the best of our knowledge, there is no published study that compares native and sterile soils in the scope of the growth-promoting and stress-alleviating effects of *S. indica* under B toxic conditions or any other stress conditions. However, a study conducted to investigate the effect of arbuscular mycorrhiza on wheat plants grown in non-autoclaved and autoclaved soils showed that the shoot dry weights of wheat plants were significantly increased by soil sterilization [109]. Moreover, experimental input of Gholami et al. [110] showed that the effect of *Azospirillum brasilense* (DSM 1690) on the growth of maize was significantly more pronounced in sterile soil compared to native soil.

Chlorosis and necrosis on older leaves are typical symptoms of plants affected by B toxicity. After being loaded into xylem, B is transported to the shoot by the transpiration stream, which is necessary for the long-distance translocation of solutes in the xylem [4,111]. Thus, B accumulates primarily in tissues with high transpiration rates. Along the transpiration stream, B is transported and accumulates at the end of xylem vessels. As a result, the element is significantly more concentrated in leaf tips and margins compared to the rest of the leaf [112,113]. It is well documented that B can also be transported via the phloem in some species but not others [4,113,114]. Species translocating mainly sucrose in their phloem, including wheat, barley, canola, and walnut, display extremely low B mobility in the phloem. Therefore, older leaves of these species have relatively higher B concentrations. The current work revealed that the leaves of both cultivars turned chlorotic and necrotic under B toxic conditions. In addition, these symptoms were more prominent in older leaves, and increased B toxicity gradually exacerbated these symptoms, as expected (Appendix A).

In this study, the leaf MDA concentrations and REL were significantly reduced by *S. indica* inoculation whereas toxic B levels resulted in elevated levels of MDA and REL in both cultivars (Figure 3 and Figure 4). Therefore, it can be concluded that the membrane integrity of plant leaves was disrupted by B toxicity and the ameliorative effects of *S. indica* appeared in both cultivars. A study showed that B-toxicity reduced growth and increased the amount of H_2_O_2_ and MDA in root and shoot tissues of wheat cultivars [98]. Similarly, Karabal et al. [29] reported elevated MDA concentrations and REL in a dose-dependent manner in leaves of sensitive and tolerant barley cultivars (*Hordeum vulgare* L.) subjected to B toxicity. The link between *S. indica* and MDA concentration under stress conditions has been the subject of numerous studies to date. For example, *S. indica* inoculation caused an increase in proline content and a significant reduction in MDA accumulation in corn (*Zea mays* L.) and rice (*Oryza sativa* L.) plants, respectively [61,115]. Furthermore, studies reported that *S. indica* reduced REL under drought stress, thereby re-confirming the importance of this fungus in preventing oxidative damage to membranes, induced by ROS [88,116].

In the chloroplast, a decrease in chlorophyll content causes structural changes, which diminish photosynthetic performance [117]. An excess of B exerts different effects on plants, such as a reduction in leaf chlorophyll contents and photosynthetic rates [14,16]; therefore, the reductions in chlorophyll concentration under B toxicity can be regarded as a biological index to determine plant tolerance to B stress. As shown in Table 3, in durum wheat, improved chlorophyll levels were observed in the inoculated group whereas, in Nusrat, inoculation did not result in any considerable change. This effect is likely based on differences between the two genotypes in terms of responsiveness to *S. indica* and B toxicity tolerance. As reported by Yau et al. [7], durum wheat is highly sensitive to B toxicity and durum wheat cultivars were reported to be more sensitive than most bread wheat cultivars [81]. According to Ogunwole et al. [118], toxic B levels resulted in significantly reduced chlorophyll a, b, and total chlorophyll concentrations, and according to their interpretations, this problem could be caused by oxidative injuries which led to higher chlorophyll degradation or inhibited chlorophyll synthesis. Moreover, an increase in the MDA and H_2_O_2_ contents of plants subjected to B toxicity was earlier reported in chickpea (*Cicer arietinum* L.) and tomato (*Solanum lycopersicum* L.) plants [23,25,119]. These findings were attributed to oxidative stress and membrane peroxidation which may enhance chlorophyll degradation and reduced chlorophyll synthesis. The aforementioned findings of MDA and REL are in line with our findings as explained above. Moreover, a study conducted on rice plants (*Oryza sativa* L.) to investigate the effect of *S. indica* infection under salt stress conditions showed that *S. indica* colonization significantly enhanced Chl a, Chl b, and carotenoid content as compared to corresponding control plants [63]. Various studies have also showed that photosynthetic improvements, including, but not limited to, improvement in chlorophyll levels and the photochemical efficiency of photosystem II, can be responsible for the observed positive effects of *S. indica* under salinity stress [54,120,121].

While B is an essential micronutrient for all seed plants, its adequacy range is particularly narrow and, therefore, excess B can easily result in toxic effects [37]. This study revealed that *S. indica* did not cause any significant effect on wheat B uptake, while soil sterilization significantly reduced the B uptake (Figure 5). As expected, shoot B concentrations increased with increasing B supply. In studies which focused on the toxic effects of other elements, *S. indica* symbiosis was associated with reduced tissue concentrations of the toxic elements. A study conducted on tomatoes (*Solanum lycopersicum* L.) by Ghorbani et al. [54] revealed that *S. indica* inoculation reduced absorption and transport of Na^+^ to the shoot compared to non-inoculated plants under salinity. *S. indica* was also reported to diminish tissue levels of heavy metals [122]. According to another study, *S. indica* immobilized Cd in roots and reduced Cd concentrations in stems and leaves of sunflower plants under Cd toxic conditions [123]. The ability to sustain low concentrations of B in plant tissues appears to be a primary mechanism that is partially attributed to the active efflux of B from the roots and this primary mechanism is similar for all species tested [74,124]. As a result, in general, B toxicity-tolerant genotypes accumulate less B than sensitive genotypes, indicating that exclusion rather than internal tolerance mechanisms are often in charge [124]. However, in the present study, even though shoot B concentrations were not reduced by *S. indica*, different mechanisms such as antioxidant defense were apparently activated to ameliorate B toxicity and thereby maintain growth and development.

Plants obtain P from the environment directly by their roots or indirectly through symbiotic associations such as those with mycorrhizal fungi [20,125]. The current study showed that shoot P concentrations were reduced by *S. indica* but increased by soil sterilization. Moreover, shoot P concentrations increased with increasing B treatments (Figure 7), possibly due to ‘concentration’ effect. Previous studies have suggested that *S. indica*, especially in arid and semi-arid regions, may help with P acquisition through mechanisms involving a phosphate transporter (PiPT) and an acid phosphatase [126,127]. In such studies, growth promotion by *S. indica* has been at least partly attributed to enhanced P uptake [128]. Here, contrary to these findings, *S. indica* did not increase P concentrations in wheat but enhancement of plant growth by *S. indica* inoculation should also be taken into account (Table 4 and Figure 7). During the vegetative stage of growth, the P requirement for optimal growth is in the range between 0.3 and 0.5% dry weight [20,129], and the measured P concentrations in this study were within this range. It is possible that *S. indica* inoculation “diluted” P in the above-ground parts of plants by promoting shoot growth. When the P contents of the cultivars are calculated, it appears that the P contents were improved by *S. indica* significantly. 

Among the essential mineral concentration measured in plant samples in this study, Mn concentrations were affected to the greatest extent by soil sterilization, and, therefore, in addition to B and P concentrations, Mn concentrations were also reported (Figure 6). Soil sterilization may lead to Mn toxicity via killing the Mn-oxidizing soil bacteria that are responsible converting plant-available Mn^2+^ into unavailable higher oxides [130,131]. In this study, soil sterilization resulted in a three-fold enhancement of Mn concentrations (Figure 6). Similarly, Miransari et al. [132] conducted a study to investigate the effects of arbuscular mycorrhiza, soil sterilization, and soil compaction on wheat (*Triticum aestivum* L.) nutrient uptake and reported that soil sterilization significantly increased Mn uptake. In the present study, *S. indica* significantly reduced shoot Mn concentrations in Nusrat but not in Saricanak-98 (Figure 6). So, it appeared that the effect of *S. indica* on Mn uptake is genotype-dependent and *S. indica* could help alleviate Mn toxicity, depending on the conditions. In bread wheat cultivar, B toxicity was also associated with higher Mn accumulation. Excess B can promote Mn accumulation in plants [133,134,135]. So, higher shoot Mn accumulation can be attributed to toxic B levels in Nusrat.

In the current investigation, significant *S. indica*-mediated increase in grain number, straw dry weight and grain yield were monitored for both cultivars (Figure 9, Figure 10 and Figure 11). Consistent with our data, *S. indica*-mediated yield increases were reported in wheat [66] and barley [136].

However, the number of studies in which cereals (i.e., wheat and barley) were grown to full maturity in the presence or absence of symbiosis with *S. indica* and significant yield increases were reported are limited because most studies published so far dealt only with effects observed at the vegetative stage [59,137] and none of those studies reported yield effects that were dependent on B toxicity. Apart from that, irrespective of *S. indica*, those yield parameters were increased by soil sterilization probably thanks to the elimination of soil-borne pathogens as explained above. Therefore, soil sterilization had significant effects on the shoot growth as well as yield parameters, indicating that the native soil was a significant source of biotic stress. Excessive B restricts the production of starch or creates B-carbohydrate complexes resulting in delayed grain formation [35] and because of the negative effects of excessive B on pollen development and pollen tube growth, seed and fruit set may be hampered [138]. However, the straw dry matter was not affected by B toxicity in both cultivars (Figure 10). According to Serfling et al. [57], even though *S. indica* improved the straw biomass of wheat, improvement in grain yield was not statistically significant. In the current study, straw biomass was increased significantly by *S. indica*; however, grain yield did not (Figure 10 and Figure 11). As a result, when grain yield was restricted as happened in this study, the straw dry weight can remain relatively high. In Nusrat, the straw B concentration was reduced by *S. indica*, but the grain B concentration was not affected whereas in Saricanak-98 the straw B concentration was reduced but the grain B concentration was not (Table 6 and Table 7). It appeared that the translocation of B was affected differently by *S. indica* in both cultivars. In addition to that, grain B levels were negligibly small when compared to straw B concentrations, which makes sense in the light of the limited phloem mobility of B in wheat [139].

## 5. Conclusions

*Serendipita indica* has substantial potential as a plant growth-promoting micro-organism. This study provided insight into the extent and mechanisms of the contribution of *S. indica* to bread as well as durum wheat in terms of growth and yield parameters in the presence or absence of B toxicity as an abiotic stress factor which had previously not been investigated in relation to *S. indica*. It was documented for the first time that wheat growth and yield parameters can be significantly improved by *S. indica* under B toxicity although the extent of the benefit provided by *S. indica* is not proportional to stress severity. Soil sterilization was not required to observe the positive effects of *S. indica*, indicating that *S. indica* was not suppressed by the native soil microflora. Inoculation with *S. indica* helped wheat plants by alleviating membrane damage but not necessarily by reducing toxic B concentrations in the shoot or increasing the concentrations of potentially limiting essential minerals such as P. Since *S. indica* can be axenically produced on an industrial scale, we recommend the consideration of this endophyte as a biofertilizer for sustainable agriculture under adverse soil conditions, such as B toxicity. More research is, however, needed to evaluate the potential of *S. indica* under field conditions.

## Figures and Tables

**Figure 1 biology-12-01098-f001:**
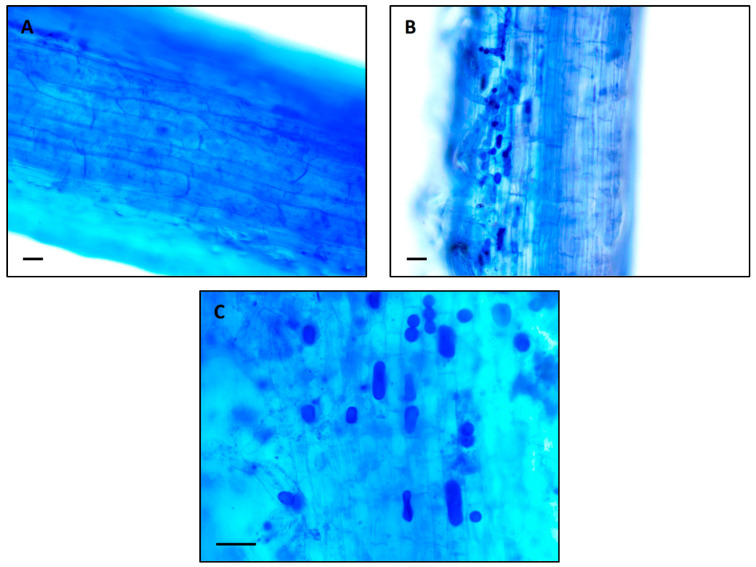
Trypan blue staining of (**A**) control and (**B**,**C**) inoculated durum wheat (*Triticum durum* cv. Saricanak-98) roots at medium B toxicity (20 mg kg^−1^) showing intracellular *S. indica* spores 41 days after sowing. Bars (black lines) represent 50 µm.

**Figure 2 biology-12-01098-f002:**
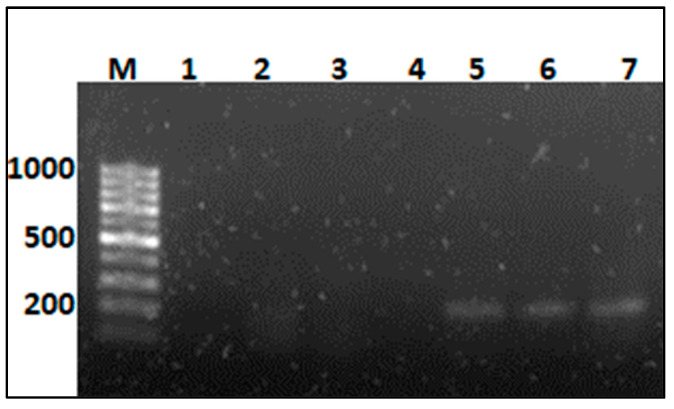
Polymerase chain reaction (PCR) detection of EF-1α (*tef*) gene of *S. indica* in DNA samples isolated from wheat roots 41 days after sowing (M: 100 bp DNA ladder; lane 1: negative control; lane 2: non-inoculated durum wheat roots at high B toxicity; lane 3: non-inoculated bread wheat roots at high B toxicity; lane 4: non-inoculated bread wheat roots at medium B toxicity; lane 5: inoculated bread wheat roots at medium B toxicity; lane 6: inoculated bread wheat roots at high B toxicity; and lane 7: inoculated durum wheat roots at high B toxicity).

**Figure 3 biology-12-01098-f003:**
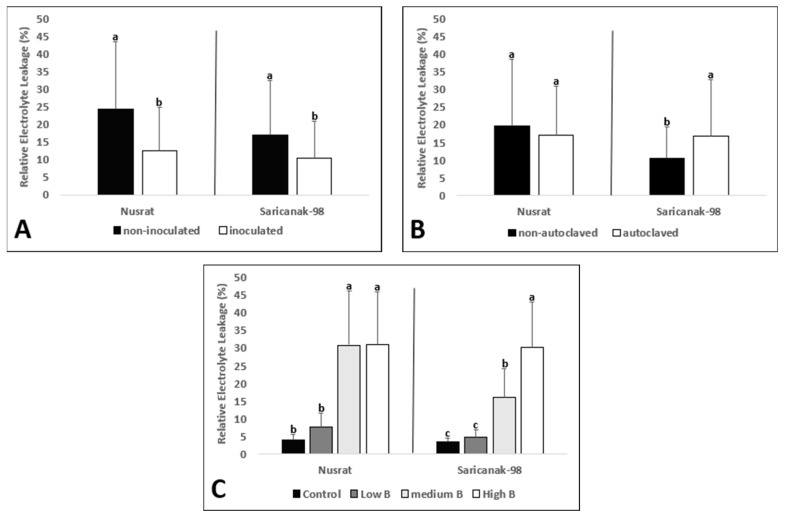
Relative electrolyte leakage (%) from the 2nd oldest leaves of 41-day-old bread wheat (*Triticum aestivum* cv. Nusrat) and durum wheat (*Triticum durum* cv. Saricanak-98) plants as affected by *S. indica* (**A**), soil sterilization (**B**) (autoclaved or non-autoclaved soil) and B toxicity (**C**) (control: 1 mg kg^−1^; low B toxicity: 10 mg kg^−1^; medium B toxicity: 20 mg kg^−1^; high B toxicity: 30 mg kg^−1^) under greenhouse conditions (at 41 DAS). Different letters indicate significant differences between means according to Tukey’s HSD test (*p* ≤ 0.05). Statistical analysis was performed separately for Nusrat and Saricanak-98. Based on the ANOVA results reported in Table 1, this figure focuses on the ‘main effects’ of the factors (inoculation, sterilization, and B toxicity) on the REL. In each graph, the reported mean values were averaged over the other factors. Accordingly, in parts A, B, and C, each column represents the mean values obtained from 32, 32, and 16 pots, respectively.

**Figure 4 biology-12-01098-f004:**
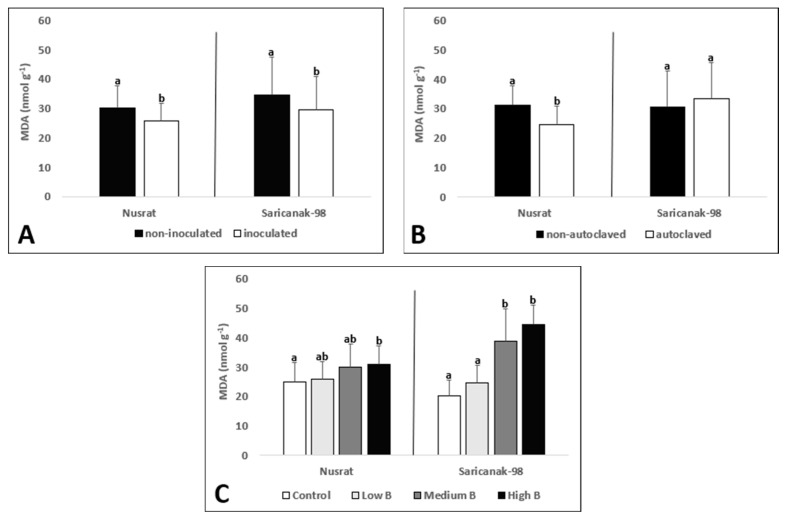
MDA concentration (nmol g^−1^) of the 3rd and 4th oldest leaves (pooled) of 41-day-old bread wheat (*Triticum aestivum* cv. Nusrat) and durum wheat (*Triticum durum* cv. Saricanak-98) plants as affected by *S. indica* (**A**), soil sterilization (**B**) (autoclaved or non-autoclaved soil) and B toxicity (**C**) (control: 1 mg kg^−1^; low B toxicity: 10 mg kg^−1^; medium B toxicity: 20 mg kg^−1^; high B toxicity: 30 mg kg^−1^) under greenhouse conditions (at 41 DAS). Different letters indicate significant differences between means according to Tukey’s HSD test (*p* ≤ 0.05). Statistical analysis was performed separately for Nusrat and Saricanak-98. Based on the ANOVA results reported in Table 1, this figure focuses on the ‘main effects’ of the factors (inoculation, sterilization, and B toxicity) on the MDA concentrations. In each graph, the reported mean values were averaged over the other factors. Accordingly, in parts A, B, and C, each column represents the mean values obtained from 32, 32, and 16 pots, respectively.

**Figure 5 biology-12-01098-f005:**
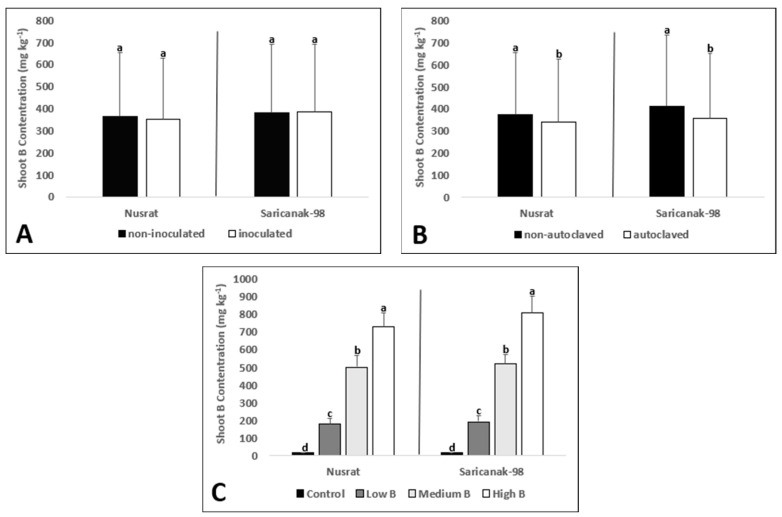
Shoot B concentration (mg kg^−1^) of 41-day-old bread wheat (*Triticum aestivum* cv. Nusrat) and durum wheat (*Triticum durum* cv. Saricanak-98) plants as affected by *S. indica* (**A**), soil sterilization (**B**) (autoclaved or non-autoclaved soil) and B toxicity (**C**) (control: 1 mg kg^−1^; low B toxicity: 10 mg kg^−1^; medium B toxicity: 20 mg kg^−1^; high B toxicity: 30 mg kg^−1^) under greenhouse conditions (at 41 DAS). Different letters indicate significant differences between means according to Tukey’s HSD test (*p* ≤ 0.05). Statistical analysis was performed separately for Nusrat and Saricanak-98. Based on the ANOVA results reported in Table 4, this figure focuses on the ‘main effects’ of the factors (inoculation, sterilization, and B toxicity) on the shoot B concentrations. In each graph, the reported mean values were averaged over the other factors. Accordingly, in parts A, B, and C, each column represents the mean values obtained from 32, 32, and 16 pots, respectively.

**Figure 6 biology-12-01098-f006:**
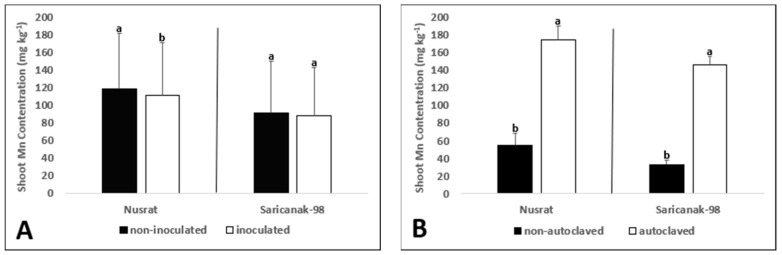
Shoot Mn concentration (mg kg^−1^) of 41-day-old bread wheat (*Triticum aestivum* cv. Nusrat) and durum wheat (*Triticum durum* cv. Saricanak-98) plants as affected by *S. indica* (**A**), soil sterilization (**B**) (autoclaved or non-autoclaved soil) and B toxicity (**C**) (control: 1 mg kg^−1^; low B toxicity: 10 mg kg^−1^; medium B toxicity: 20 mg kg^−1^; high B toxicity: 30 mg kg^−1^) under greenhouse conditions (at 41 DAS). Different letters indicate significant differences between means according to Tukey’s HSD test (*p* ≤ 0.05). Statistical analysis was performed separately for Nusrat and Saricanak-98. Based on the ANOVA results reported in Table 4, this figure focuses on the ‘main effects’ of the factors (inoculation, sterilization, and B toxicity) on the shoot Mn concentrations. In each graph, the reported mean values were averaged over the other factors. Accordingly, in parts A, B, and C, each column represents the mean values obtained from 32, 32, and 16 pots, respectively.

**Figure 7 biology-12-01098-f007:**
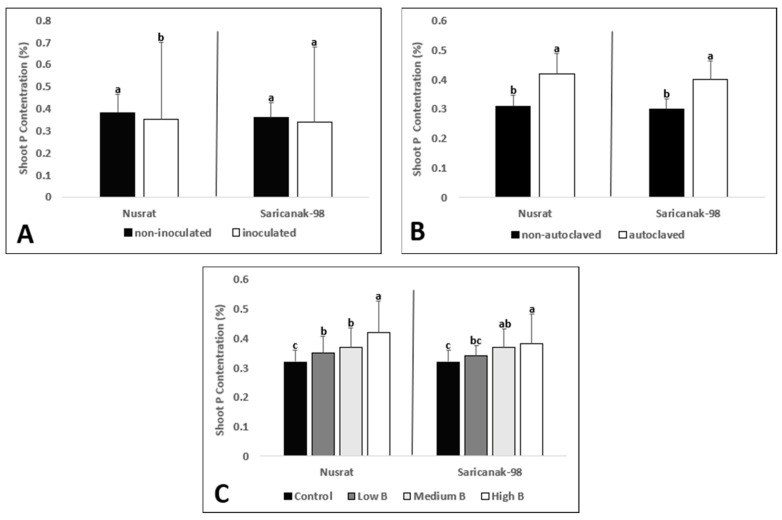
Shoot P concentration (%) of 41-day-old bread wheat (*Triticum aestivum* cv. Nusrat) and durum wheat (*Triticum durum* cv. Saricanak-98) plants as affected by *S. indica* (**A**), soil sterilization (**B**) (autoclaved or non-autoclaved soil) and B toxicity (**C**) (control: 1 mg kg^−1^; low B toxicity: 10 mg kg^−1^; medium B toxicity: 20 mg kg^−1^; high B toxicity: 30 mg kg^−1^) under greenhouse conditions (at 41 DAS). Different letters indicate significant differences between means according to Tukey’s HSD test (*p* ≤ 0.05). Statistical analysis was performed separately for Nusrat and Saricanak-98. Based on the ANOVA results reported in Table 4, this figure focuses on the ‘main effects’ of the factors (inoculation, sterilization, and B toxicity) on the shoot P concentrations. In each graph, the reported mean values were averaged over the other factors. Accordingly, in parts A, B, and C, each column represents the mean values obtained from 32, 32, and 16 pots, respectively.

**Figure 8 biology-12-01098-f008:**
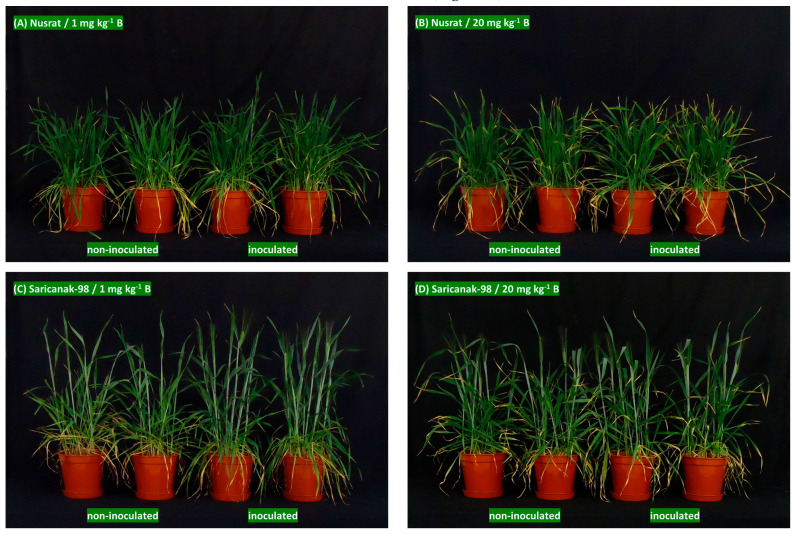
Effects of *S. indica* on 50-day-old bread wheat (*Triticum aestivum* cv. Nusrat) (**A**,**B**) and durum wheat (*Triticum durum* cv. Saricanak-98) (**C**,**D**) grown in non-autoclaved soil at control B (1 mg kg^−1^) and severe B toxicity (20 mg kg^−1^) levels under greenhouse conditions (Experiment 2, the photos were taken at 50 DAS).

**Figure 9 biology-12-01098-f009:**
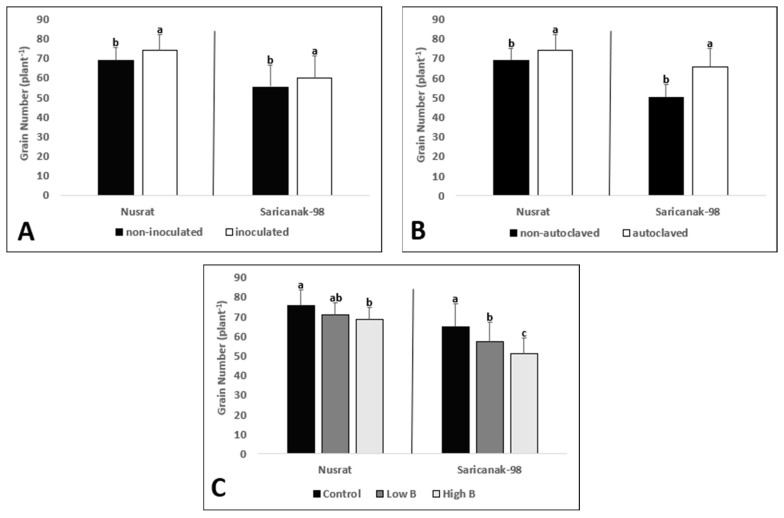
Grain number (plant^−1^) of mature bread wheat (*Triticum aestivum* cv. Nusrat) and durum wheat (*Triticum durum* cv. Saricanak-98) plants as affected by *S. indica* (**A**), soil sterilization (**B**) (autoclaved or non-autoclaved soil) and B toxicity (**C**) (control: 1 mg kg^−1^; low B toxicity: 10 mg kg^−1^; high B toxicity: 20 mg kg^−1^) under greenhouse conditions (at 111 DAS). Different letters indicate significant differences between means according to Tukey’s HSD test (*p* ≤ 0.05). Statistical analysis was performed separately for Nusrat and Saricanak-98. Based on the ANOVA results reported in Table 5, this figure focuses on the ‘main effects’ of the factors (inoculation, sterilization, and B toxicity) on the grain numbers. In each graph, the reported mean values were averaged over the other factors. Accordingly, in parts A, B, and C, each column represents the mean values obtained from 32, 32, and 16 pots, respectively.

**Figure 10 biology-12-01098-f010:**
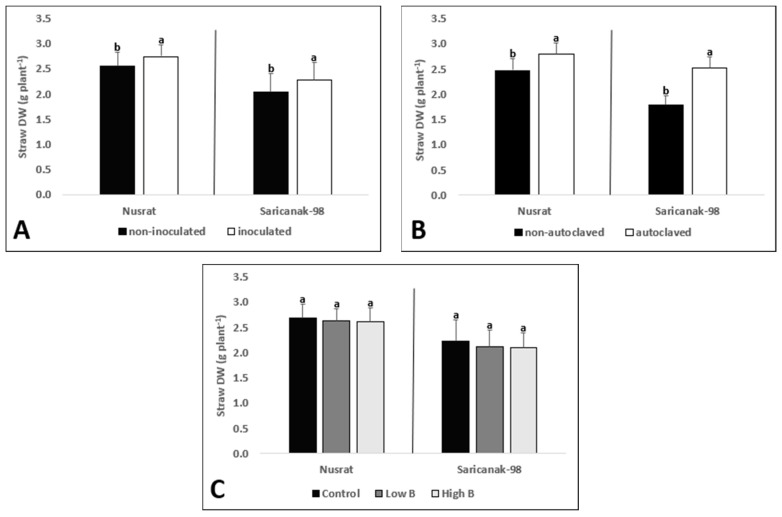
Straw dry weight (g plant^−1^) of mature bread wheat (*Triticum aestivum* cv. Nusrat) and durum wheat (*Triticum durum* cv. Saricanak-98) plants as affected by *S. indica* (**A**), soil sterilization (**B**) (autoclaved or non-autoclaved soil) and B toxicity (**C**) (control: 1 mg kg^−1^; low B toxicity: 10 mg kg^−1^; high B toxicity: 20 mg kg^−1^) under greenhouse conditions (at 41 DAS). Different letters indicate significant differences between means according to Tukey’s HSD test (*p* ≤ 0.05). Statistical analysis was performed separately for Nusrat and Saricanak-98. Based on the ANOVA results reported in Table 5, this figure focuses on the ‘main effects’ of the factors (inoculation, sterilization, and B toxicity) on the straw dry weights. In each graph, the reported mean values were averaged over the other factors. Accordingly, in parts A, B, and C, each column represents the mean values obtained from 32, 32, and 16 pots, respectively.

**Figure 11 biology-12-01098-f011:**
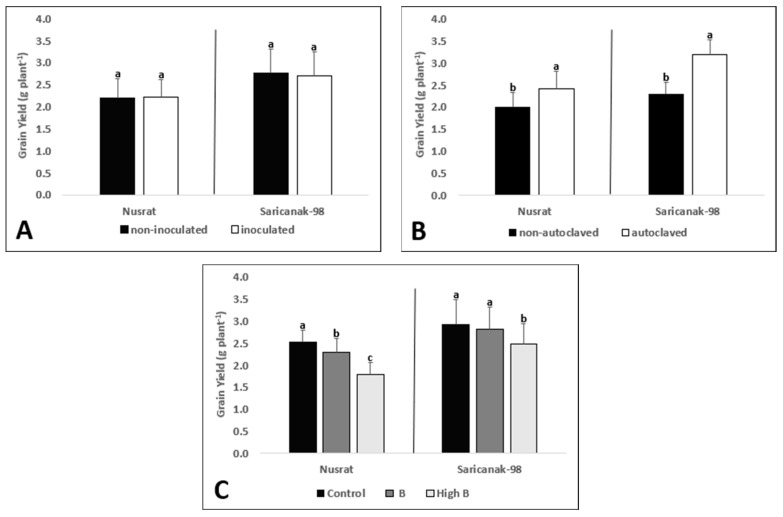
Grain yield (g plant^−1^) of mature bread wheat (*Triticum aestivum* cv. Nusrat) and durum wheat (*Triticum durum* cv. Saricanak-98) plants as affected by *S. indica* (**A**), soil sterilization (**B**) (autoclaved or non-autoclaved soil) and B toxicity (**C**) (control: 1 mg kg^−1^; low B toxicity: 10 mg kg^−1^; high B toxicity: 20 mg kg^−1^) under greenhouse conditions (at 41 DAS). Different letters indicate significant differences between means according to Tukey’s HSD test (*p* ≤ 0.05). Statistical analysis was performed separately for Nusrat and Saricanak-98. Based on the ANOVA results reported in Table 5, this figure focuses on the ‘main effects’ of the factors (inoculation, sterilization, and B toxicity) on the grain yields. In each graph, the reported mean values were averaged over the other factors. Accordingly, in parts A, B, and C, each column represents the mean values obtained from 32, 32, and 16 pots, respectively.

**Table 1 biology-12-01098-t001:** Three-way analysis of variance (ANOVA) of the effects of inoculation, soil sterilization and boron applications, as well as their interactions on shoot DW (g), REL (%), MDA (nmol g^−1^), and total chlorophyll (nmol g^−1^) of bread wheat (*Triticum aestivum* cv. Nusrat) and durum wheat (*Triticum durum* cv. Saricanak-98) grown under greenhouse conditions of both vegetative and yield stages (41 and 111 days after sowing): DF (degrees of freedom), F Pr. (F value probabilities) and Tukey’s protected HSD_0.05_ test scores.

	Source of Variation	DF	Shoot DW (g)		REL (%)		MDA (nmol g^−1^)		Total Chlorophyll (nmol g^−1^)	
			F Pr.	HSD_0.05_	F Pr.	HSD_0.05_	F Pr.	HSD_0.05_	F Pr.	HSD_0.05_
Nusrat	Inoculation (I)	1	<0.001	0.036	<0.001	4.1	0.006	3.2	0.697	*n.s.*
	Soil sterilization (S)	1	<0.001	0.036	0.213	*n.s.*	<0.001	3.2	0.009	83
	Boron supply (B)	3	<0.001	0.066	<0.001	7.7	0.020	5.8	0.075	*n.s.*
	I X S	1	0.123	*n.s.*	0.060	*n.s.*	0.463	*n.s.*	0.004	156
	I X B	3	0.652	*n.s.*	0.005	12.9	0.436	*n.s.*	0.156	*n.s.*
	S X B	3	0.001	0.114	0.023	12.9	0.678	*n.s.*	0.023	262
	I X S X B	3	0.018	0.184	0.234	*n.s.*	0.925	*n.s.*	0.349	*n.s.*
Saricanak-98	Inoculation (I)	1	<0.001	0.024	<0.001	2.5	0.008	3.8	0.014	76
	Soil sterilization (S)	1	<0.001	0.024	<0.001	2.5	0.163	*n.s.*	0.467	*n.s.*
	Boron supply (B)	3	<0.001	0.045	<0.001	4.6	<0.001	7.2	0.030	143
	I X S	1	0.037	0.045	0.006	4.6	0.451	*n.s.*	0.405	*n.s.*
	I X B	3	0.361	*n.s.*	0.001	7.8	0.487	*n.s.*	0.005	241
	S X B	3	0.497	*n.s.*	<0.001	7.8	0.608	*n.s.*	0.371	*n.s.*
	I X S X B	3	0.869	*n.s.*	0.201	*n.s.*	0.519	*n.s.*	0.437	*n.s.*

**Table 2 biology-12-01098-t002:** Effects of *S. indica* on the shoot dry weights (g plant ^−1^) of 41-day-old bread wheat (*Triticum aestivum* cv. Nusrat) and durum wheat (*Triticum durum* cv. Saricanak-98) plants grown in autoclaved or non-autoclaved soil supplemented with different concentration of B in the form of H_3_BO_3_ (control: 1 mg kg^−1^; low B toxicity: 10 mg kg^−1^; medium B toxicity: 20 mg kg^−1^; high B toxicity: 30 mg kg^−1^) under greenhouse conditions (at 41 DAS). Data points represent means and standard deviations of four replicates.

Shoot DW (g Plant ^−1^)						
Cultivar	Soil Sterilization	*S. indica*	B Supply (mg kg^−1^)			
			1	10	20	30
**Nusrat**	**Autoclaved**	**(−)**	0.58 ± 0.08	0.53 ± 0.17	0.56 ± 0.11	0.47 ± 0.06
		**(+)**	0.73 ± 0.08	0.82 ± 0.07	0.68 ± 0.02	0.58 ± 0.03
	**Non-autoclaved**	**(−)**	0.62 ± 0.10	0.50 ± 0.03	0.33 ± 0.03	0.27 ± 0.03
		**(+)**	0.71 ± 0.03	0.54 ± 0.08	0.52 ± 0.04	0.38 ± 0.02
**Saricanak-98**	**Autoclaved**	**(−)**	0.68 ± 0.05	0.67 ± 0.03	0.56 ± 0.04	0.50 ± 0.05
		**(+)**	0.84 ± 0.03	0.76 ± 0.10	0.63 ± 0.04	0.58 ± 0.03
	**Non-autoclaved**	**(−)**	0.53 ± 0.02	0.52 ± 0.08	0.35 ± 0.02	0.34 ± 0.03
		**(+)**	0.60 ± 0.06	0.57 ± 0.06	0.39 ± 0.06	0.36 ± 0.04

**Table 3 biology-12-01098-t003:** Effects of *S. indica* on the total chlorophyll (nmol g^−1^) of 41-day-old bread wheat (*Triticum aestivum* cv. Nusrat) and durum wheat (*Triticum durum* cv. Saricanak-98) plants grown in autoclaved or non-autoclaved soil supplemented with different concentration of B in the form of H_3_BO_3_ (control: 1 mg kg^−1^; low B toxicity: 10 mg kg^−1^; medium B toxicity: 20 mg kg^−1^; high B toxicity: 30 mg kg^−1^) under greenhouse conditions (at 41 DAS). Data points represent means and standard deviations of four replicates.

Total Chlorophyll (nmol g^−1^)						
Cultivar	Soil Sterilization	*S. indica*	B Supply (mg kg^−1^)			
			1	10	20	30
**Nusrat**	**Autoclaved**	**(−)**	1067 ± 104	794 ± 147	658 ± 72	529 ± 50
		**(+)**	1155 ± 127	879 ± 40	734 ± 150	778 ± 46
	**Non-autoclaved**	**(−)**	873 ± 213	897 ± 227	616 ± 118	507 ± 179
		**(+)**	700 ± 169	643 ± 44	927 ± 253	727 ± 86
**Saricanak-98**	**Autoclaved**	**(−)**	723 ± 162	789 ± 169	842 ± 126	778 ± 167
		**(+)**	1092 ± 97	808 ± 40	878 ± 230	679 ± 92
	**Non-autoclaved**	**(−)**	822 ± 120	832 ± 327	659 ± 72	786 ± 301
		**(+)**	795 ± 245	853 ± 126	828 ± 129	604 ± 70

**Table 4 biology-12-01098-t004:** Three-way analysis of variance (ANOVA) of the effects of inoculation, soil sterilization and boron applications as well as their interactions on shoot concentrations of B (mg kg^−1^), Mn (mg kg^−1^), P(%) and straw B (mg kg^−1^), of bread wheat (*Triticum aestivum* cv. Nusrat) and durum wheat (*Triticum durum* cv. Saricanak-98) grown under greenhouse conditions of both vegetative and yield stages (41 and 111 DAS): DF (degrees of freedom), F Pr. (F value probabilities) and Tukey’s protected HSD_0.05_ test scores.

	Source of Variation	DF	Shoot B Conc. (mg kg^−1^)		Shoot Mn Conc. (mg kg^−1^)		Shoot P Conc. (%)		DF	Straw B Conc. (mg kg^−1^)	
			F Pr.	HSD_0.05_	F Pr.	HSD_0.05_	F Pr.	HSD_0.05_		F Pr.	HSD_0.05_
Nusrat	Inoculation (I)	1	0.357	*n.s.*	0.004	5.26	<0.001	0.014	1	0.0012	19
	Soil sterilization (S)	1	0.015	26	<0.001	5.26	<0.001	0.014	1	<0.001	19
	Boron supply (B)	3	<0.001	48	<0.001	9.75	<0.001	0.026	2	<0.001	28
	I X S	1	0.885	*n.s.*	0.266	*n.s.*	0.478	*n.s.*	1	0.534	*n.s.*
	I X B	3	0.512	*n.s.*	0.885	*n.s.*	0.447	*n.s.*	2	0.188	*n.s.*
	S X B	3	0.040	80	0.002	16.42	<0.001	0.047	2	<0.001	48
	I X S X B	3	0.424	*n.s.*	0.985	*n.s.*	0.914	*n.s.*	2	0.25	*n.s.*
Saricanak-98	Inoculation (I)	1	0.826	*n.s.*	0.132	*n.s.*	0.051	*n.s.*	1	0.250	*n.s.*
	Soil sterilization (S)	1	<0.001	27	<0.001	3.31	<0.001	0.016	1	0.002	31
	Boron supply (B)	3	<0.001	51	0.735	*n.s.*	<0.001	0.029	2	<0.001	47
	I X S	1	0.373	*n.s.*	0.002	6.2	0.733	*n.s.*	1	0.568	*n.s.*
	I X B	3	0.997	*n.s.*	0.176	*n.s.*	0.134	*n.s.*	2	0.075	*n.s.*
	S X B	3	0.123	*n.s.*	0.005	10.44	<0.001	0.05	2	0.154	*n.s.*
	I X S X B	3	0.908	*n.s.*	0.531	*n.s.*	0.774	*n.s.*	2	0.747	*n.s.*

**Table 5 biology-12-01098-t005:** Three-way analysis of variance (ANOVA) of the effects of inoculation, soil sterilization and boron applications, as well as their interactions on grain B concentrations (mg kg^−1^), straw DW (g), grain number (plant^−1^) and grain yield (g) of bread wheat (*Triticum aestivum* cv. Nusrat) and durum wheat (*Triticum durum* cv. Saricanak-98) grown under greenhouse conditions of both vegetative and yield stages (41 and 111 DAS): DF (degrees of freedom), F Pr. (F value probabilities) and Tukey’s protected HSD_0.05_ test scores.

	Source of Variation	DF	Grain B Conc. (mg kg^−1^)		Straw DW (g)		Grain Number (Plant^−1^)		Grain Yield Plant^−1^ (g)	
**Nusrat**			F Pr.	HSD_0.05_	F Pr.	HSD_0.05_	F Pr.	HSD_0.05_	F Pr.	HSD_0.05_
	**Inoculation (I)**	1	0.220	*n.s.*	0.001	0.117	0.003	3.463	0.595	*n.s.*
	**Soil sterilization (S)**	1	0.066	*n.s.*	<0.001	0.117	0.004	3.463	<0.001	0.125
	**Boron supply (B)**	2	<0.001	0.212	0.567	*n.s.*	0.006	5.110	<0.001	0.185
	**I X S**	1	0.163	*n.s.*	0.120	*n.s.*	0.363	*n.s.*	0.145	*n.s.*
	**I X B**	2	0.244	*n.s.*	0.207	*n.s.*	0.033	8.897	0.342	*n.s.*
	**S X B**	2	0.013	0.372	0.523	*n.s.*	0.379	*n.s.*	0.250	*n.s.*
	**I X S X B**	2	0.340	*n.s.*	0.977	*n.s.*	0.181	*n.s.*	0.970	*n.s.*
**Saricanak-98**	**Inoculation (I)**	1	0.022	0.412	0.041	0.231	0.011	3.463	0.385	*n.s.*
	**Soil sterilization (S)**	1	0.001	0.412	<0.001	0.231	<0.001	3.463	<0.001	0.150
	**Boron supply (B)**	2	<0.001	0.611	0.608	*n.s.*	<0.001	5.112	<0.001	0.219
	**I X S**	1	0.643	*n.s.*	0.410	*n.s.*	0.719	*n.s.*	0.889	*n.s.*
	**I X B**	2	0.162	*n.s.*	0.462	*n.s.*	0.341	*n.s.*	0.402	*n.s.*
	**S X B**	2	0.021	1.045	0.602	*n.s.*	0.124	*n.s.*	0.648	*n.s.*
	**I X S X B**	2	0.840	*n.s.*	0.308	*n.s.*	0.497	*n.s.*	0.682	*n.s.*

**Table 6 biology-12-01098-t006:** Effects of *S. indica* on the grain B concentrations (mg kg^−1^) of mature bread wheat (*Triticum aestivum* cv. Nusrat) and durum wheat (*Triticum durum* cv. Saricanak-98) plants grown in autoclaved or non-autoclaved soil supplemented with different concentration of B in the form of H_3_BO_3_ (control: 1 mg kg^−1^; low B toxicity: 10 mg kg^−1^; high B toxicity: 20 mg kg^−1^) under greenhouse conditions (at 111 DAS). Data points represent the means and standard deviations of four replicates.

Grain B Conc. (mg kg^−1^)					
Cultivar	Soil Sterilization	*S. indica*	B Supply (mg kg^−1^)		
			1	10	20
**Nusrat**	**Autoclaved**	**(−)**	1.13 ± 0.13	1.42 ± 0.05	2.33 ± 0.13
		**(+)**	1.08 ± 0.20	1.37 ± 0.08	2.46 ± 0.52
	**Non-autoclaved**	**(−)**	0.92 ± 0.10	1.87 ± 0.27	2.80 ± 0.18
		**(+)**	0.94 ± 0.07	1.39 ± 0.16	2.68 ± 0.23
**Saricanak-98**	**Autoclaved**	**(−)**	0.74 ± 0.05	1.94 ± 0.26	3.54 ± 0.63
		**(+)**	0.68 ± 0.03	1.65 ± 0.30	2.71 ± 0.09
	**Non-autoclaved**	**(−)**	0.73 ± 0.05	2.79 ± 0.40	5.15 ± 0.78
		**(+)**	0.80 ± 0.04	2.05 ± 0.82	4.07 ± 1.50

**Table 7 biology-12-01098-t007:** Effects of *S. indica* on the straw B concentrations (mg kg^−1^) of mature bread wheat (*Triticum aestivum* cv. Nusrat) and durum wheat (*Triticum durum* cv. Saricanak-98) plants grown in autoclaved or non-autoclaved soil supplemented with different concentration of B in the form of H_3_BO_3_ (control: 1 mg kg^−1^; low B toxicity: 10 mg kg^−1^; high B toxicity: 20 mg kg^−1^) under greenhouse conditions (at 111 DAS). Data points represent the means and standard deviations of four replicates.

Straw B Conc. (mg kg^−1^)					
Cultivar	Soil Sterilization	*S. indica*	B Supply (mg kg^−1^)		
			1	10	20
**Nusrat**	**Autoclaved**	**(−)**	25 ± 2	158 ± 11	390 ± 47
		**(+)**	26 ± 3	136 ± 23	331 ± 29
	**Non-autoclaved**	**(−)**	42 ± 10	257 ± 20	616 ± 56
		**(+)**	26 ± 5	186 ± 18	586 ± 40
**Saricanak-98**	**Autoclaved**	**(−)**	30 ± 1	166 ± 17	606 ± 25
		**(+)**	24 ± 2	181 ± 14	513 ± 53
	**Non-autoclaved**	**(−)**	42 ± 6	228 ± 24	659 ± 91
		**(+)**	29 ± 2	257 ± 84	615 ± 74

## Data Availability

All data generated are in the article.

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
