# Peer review of "The Fungal Root Endophyte *Serendipita indica* (*Piriformospora indica*) Enhances Bread and Durum Wheat Performance under Boron Toxicity at Both Vegetative and Generative Stages of Development through Mechanisms Unrelated to Mineral Homeostasis"

_biology, 2023, doi:10.3390/biology12081098_

Round 1

Reviewer 1 Report

  This manuscript presents valuable information on how endophytic fungi Piriformospora indica enhances bread and durum wheat growth under boron toxicity. The results obtained by the authors are primarily interesting from a practical perspective, and can be used in agriculture.  The manuscript is well written and the data presented supports the conclusions. The abstract, introduction and discussion are provide a clear context for the study. However, I have a number of questions and recommendations to the “Materials and methods” and “Results”. The article can be accepted for publication with minor revision.

Title

In the title, the authors use the terms early and late stages of development, however, plants were used in the experiments on days 40 and 51 after sowing. I'm guessing that 40 days is not an early stage, the 7 days  or maximum of 20 are. I think the title should be changed. 

One of the shortcomings of this article is the overload of illustrative material. I would recommend transferring Figures 1, 3, and 9 to the appendix.

Experimental Design:

Line 187,197,204

- Why were the spore concentrations in the first and second experiments different (9.65x105 and 13.8x105 respectively)?

Line 185, 203

-Why were the numbers of seedlings in pots in the first and second experiments different?

Lines 179,180

Describing the methods of soil preparation, the authors indicate that the following fertilizers were applied before the seedlings' transplantation: 250 mg N in the form of Ca(NO3)2.4H2O; 100 mg P in the form of KH2PO4; 10 mg Cl in the form of KCl. Why was in the second experiment only 150 mg of the Ca(NO3)2.4H2O in the soil added?

Lines 222-224

Please, indicate the author or authors of the methodology (not only in reference).

In the methods, it is not described how the shoot and straw dry weight were determined, although the data are given in the results.

Results:

- In my opinion, it would be better to divide Table 1 into two separate. Also, the days after sowing (DAS)should be indicated in the table caption. It would be nice to give a transcript of those abbreviations that are not given in the table itself.

- In the Table 2 all designations are displaced.

- In the Tables 3 and 4 the days after sowing (DAS)should be indicated.

-Figures 4,5,6,7,8,10,11,12

  In theExperimental Design” is wrote that plants were grown with or without P. indica inoculation in either autoclaved or non-autoclaved soil at 4 B levels [1 mg (control), 10 mg (low), 20 mg (medium) and 30 mg (high) B per kg dry soil in the form of boric acid]. From the figure captions, it is not clear which of the soil variants was used

Lines 538, 539  Please make reference to Table 2 here.

Lines 546-558  I think, logically this part will look better in the top of the results.

Lines 587-588 Please show data and significance   

Lines 596-597 The presence or absence of significant effects of B supply on the shoot Mn concentrations 596 was genotype-dependent (Fig. 7c). I believe that this phrase is incorrect, since you did not analyze genes associated with Mn accumulations.

I have no complaints about the Discussion section.

Reviewer 2 Report

I would like to send my review comments for the manuscript biology-2485223-peer-review-v1, entitled "The Fungal Root Endophyte Piriformospora indica Enhances Bread and Durum Wheat Growth under Boron Toxicity at both Early and Later Stages of Development through Mechanisms Unrelated to Mineral Homeostasis" for Biology. I prepared "General comments" and "Specific comments" as follows.

General comments

This research revealed functions of root endophyte Piriformospora indica and soil sterilization for boron (B) tolerance in bread wheat and durum wheat. The root endophyte could alleviate the B toxicity such as vegetative growth, straw dry weight, the grain number, and oxidative stress of both wheat cultivars. This paper describes interesting works on the effects of P. indica on B toxicity to the plant, and constructive discussion. However, the results were complicated especially in statistical analysis in figures 4–12 except for 9. The authors would need careful explanations in the results and figures captions. For instance, in Figure 7A, it was difficult to understand that P. indica significantly decreased shoot Mn concentration, because the standard deviations (SDs) seem to be large. According to replication number, statistical analysis shows significant differences regardless of the SDs. Therefore, replication numbers should be added in figure captions. Additionally, in these figure captions and/or manuscript, conditions (endophyte inoculation, soil sterilization, and B concentrations) of the representative values used in figures should be explained because the authors conducted experiments in various conditions.

Specific comments

In Experimental Design: The reason why the soil should be autoclaved as well as Line 664–666. Additionally, the reason Mn concentrations were measured should be added like Line 766–768.

Line 200: Why wasn't the B level at 30 mg kg-1 conducted?

In Table 2: The columns should be arranged as well as Table 3.

Line 336: “50 μM” should be ”50 μm”.

Line 559–560: Appropriate explanations should be added in these lines and/or figure caption. I thought the authors showed representative REL values in Figure 4. For example, Fig 4A showed the REL of wheat cultivars grown in autoclaved soil and at 1 mg kg-1 B. However, because various conditions were conducted, I did not find it clearly. Even in regard to other figures 4 –12 except for 9, appropriate explanations should be added.

Reviewer 3 Report

Dear Authors

Present manuscript entitled “The Fungal Root Endophyte Piriformospora indica Enhances Bread and Durum Wheat Growth under Boron Toxicity at both Early and Later Stages of Development through Mechanisms Unrelated to Mineral Homeostasis” discuss the adverse effects of Boron toxicity in Wheat crop; findings suggests that P. indica may be used as an effective microbial inoculant to enhance wheat growth under adverse soil conditions such as B toxicity through mechanisms that are possibly unrelated to mineral homeostasis. The research is quite interesting in reference to the endophyte and its use to mitigate the Boron toxicity; although there are certain opportunities for further improvements, please find them below.

1.       The introduction is quite long and several information may better suits to discussion part, especially from line 77-116. Please make the introduction smaller with suitable and updated information only. It is better to divide the introduction in 3 segments, first to describe the Boron toxicity and its impact, second the interaction between Wheat and endophyte, and third one is importance of the present research work.

2.       Line 131-152- The plants growth/seed germination should be in details, for example there is no information about how many lines/number of seeds per line and number of biological/technical replicates used for the study?

3.       Line 154-171- Please describe the fungal cultures growth and quantification in details; it is always better to write how you performed your experiment and later give the references. Only providing the references is not enough, and may be difficult to the readers to understand and later to perform their experiments.

4.       Line 172- the experimental design is quite complicated to understand, please keep it simpler, or better to include a diagrammatic representation.

5.       Line 222- Chlorophyll Determination need to be described; only reference is not enough.

6.       Line 225- Similarly, please include in detail.

7.       Line 254- “Genomic DNA Isolation from Plant Roots and PCR Analysis” the amplified fragment was sequenced to confirm? or it was decided just by the size?

8.       Results and discussion need to be rewritten in presentable format. The tables and figures are presented in the beginning without any description; every table and figure should be accompanied with descriptive text.

9.       Conclusion should be drawn as a short text to express the key finding.

Thank you

Regards

The manuscript need to be checked by a native speaker to rule out the minor mistakes. 

Reviewer 4 Report

Dear Authors,

I find your article very interesting and I think that it presents preliminary results which can be considered a good basis for solving problems related B toxicity in cultivated soils. The introduction is well written, and the references are appropriate, the materials and methods and discussions are also clearly stated. As for the results, in my opinion, setting results like this makes everything very confusing. I suggest the authors split the results like this:

1)Split Table 1 in three different tables. In this way Table 1 showed only Shoot DW, REL, MDA and Total Chlorophyll results.

2)      Insert the new Table 1, Table 2, Table 3 and Figures from 1 to 5 after line 579.

3)      Rename Table 1 in lines 583, 591, 599 as Table 4 (Second part of old Table 1: Shoot B Conc., Shoot Mn Conc., Shoot P Conc. Straw B. Conc.).

4)      Insert the new Table 4 and figures from 6 to 9 after line 612.

5)      Rename Table 1 in lines 614, 618, 622 and 635 as Table 5 (third part of Table 1: Grain B Conc. Straw DW, Grain number, Grain yield plant-1); rename Table 4 in line 630 as Table 6 and Table 5 in lines 637 and 637 as Table 7.

6)      Insert the new Table 5, the new Table 6, the new Table 7 and figures from 10 to 12 after line 637.

Another suggestion: as Index Fungorum and Mycobank report, the current name of the fungus is Serendipita indica. I suggest to change Piriformospora indica with S. indica. throughout the text.

Round 2

Reviewer 3 Report

Dear Authors

Thank you for providing the revised version and answering all the queries.

The manuscript has been improved significantly and i do not have any further question/suggestion.

Thank you

Regards